# Next-Generation Sequencing Screening of 43 Families with Non-Syndromic Early-Onset High Myopia: A Clinical and Genetic Study

**DOI:** 10.3390/ijms23084233

**Published:** 2022-04-11

**Authors:** Eva González-Iglesias, Ana López-Vázquez, Susana Noval, María Nieves-Moreno, María Granados-Fernández, Natalia Arruti, Irene Rosa-Pérez, Marta Pacio-Míguez, Victoria E. F. Montaño, Patricia Rodríguez-Solana, Angela del Pozo, Fernando Santos-Simarro, Elena Vallespín

**Affiliations:** 1Section of Molecular Ophthalmology, Medical and Molecular Genetics Institute (INGEMM) IdiPaz, La Paz University Hospital, 28046 Madrid, Spain; eva.gonz.igl@gmail.com (E.G.-I.); victoriaeugeniafdezmontano@hotmail.com (V.E.F.M.); patricia.rodriguez.solana@idipaz.es (P.R.-S.); 2Department of Ophthalmology, La Paz University Hospital, 28046 Madrid, Spain; analopezvazquez94@gmail.com (A.L.-V.); susana.noval@salud.madrid.org (S.N.); maria.nieves@salud.madrid.org (M.N.-M.); maria.granados@salud.madrid.org (M.G.-F.); natalia.arruti@salud.madrid.org (N.A.); irene.rosa@salud.madrid.org (I.R.-P.); 3Biomedical Research Center in the Rare Diseases Network (CIBERER), Carlos II Health Institute (ISCIII), 28029 Madrid, Spain; martapaciomiguez@gmail.com (M.P.-M.); ingemm.adelpozo@gmail.com (A.d.P.); fernando.santos@salud.madrid.org (F.S.-S.); 4Section of Neurodevelopmental Disorders, Medical and Molecular Genetics Institute (INGEMM) IdiPaz, La Paz University Hospital, 28046 Madrid, Spain; 5Section of Clinical Bioinformatics, Medical and Molecular Genetics Institute (INGEMM) IdiPaz, La Paz University Hospital, 28046 Madrid, Spain; 6Section of Clinical Genetics, Medical and Molecular Genetics Institute (INGEMM) IdiPaz, La Paz University Hospital, 28046 Madrid, Spain

**Keywords:** early-onset high myopia, next-generation sequencing, ophthalmogenetics

## Abstract

Early-onset high myopia (EoHM) is a disease that causes a spherical refraction error of ≥−6 diopters before 10 years of age, with potential multiple ocular complications. In this article, we report a clinical and genetic study of 43 families with EoHM recruited in our center. A complete ophthalmological evaluation was performed, and a sample of peripheral blood was obtained from proband and family members. DNA was analyzed using a customized next-generation sequencing panel that included 419 genes related to ophthalmological disorders with a suspected genetic cause, and genes related to EoHM pathogenesis. We detected pathogenic and likely pathogenic variants in 23.9% of the families and detected variants of unknown significance in 76.1%. Of these, 5.7% were found in genes related to non-syndromic EoHM, 48.6% in genes associated with inherited retinal dystrophies that can include a syndromic phenotype, and 45.7% in genes that are not directly related to EoHM or retinal dystrophy. We found no candidate genes in 23% of the patients, which suggests that further studies are needed. We propose a systematic genetic analysis for patients with EoHM because it helps with follow-up, prognosis and genetic counseling.

## 1. Introduction

High myopia (HM), the most challenging type, is defined as a spherical refraction error of −6 spherical diopters (SDs) or more, or an axial length greater than 26 mm. From a physiological point, myopia is a refractive error in which, in a state of relaxed accommodation, light rays parallel to the eye with an origin greater than 6 m away focus anteriorly on the retina after passing through the eye’s refractive system [1,2]. This disease is a common cause of vision loss, with uncorrected myopia the leading cause of vision impairment globally. The condition is known to be associated with ocular complications such as cataracts, glaucoma, neovascular membranes, retinal tears and retinal detachment.

The prevalence of myopia and HM worldwide is 30.6% and 2.7%, respectively [3]; however, recent studies have estimated that this prevalence will increase to 49.8% and 9.8%, respectively, by 2050 [4]. The most recent study in a Spanish population showed that the prevalence of myopia and HM in children aged 5–7 years is 20% and 3.6%, respectively [5].

The pathogenesis of myopia relies on genetic, environmental (external) and microenvironmental (internal) factors. Most children are born hyperopic, and, during the first 2 years after birth, a process known as emmetropization occurs, which is mainly influenced by a change in axial length [3]. From this moment on, environmental factors such as near work (defined as activities with a short working distance) [6], exposure to artificial light [7,8,9] can modify this condition. The onset of HM before 10 years of age, known as early-onset high myopia (EoHM), is an important aspect for research, considering that environmental factors lose relevance in children of this age, allowing an approach to the pathogenesis of high myopia as a monogenic disease [10]. Familial genetics can also contribute to this condition [11].

Physiopathological factors of the microenvironment such as oxidative stress, inflammation and angiogenesis can also induce EoHM or promote the exacerbation of the disease. Oxidative stress is produced by the metabolism of reactive species of oxygen (ROS). Retinal tissue has the highest oxygen consumption of the body and is directly exposed to natural light. These two factors produce a higher concentration of ROS in the tissue. In addition to this, EoHM patients are known to present an oxidative/antioxidative statue imbalance suggesting that oxidative stress can induce EoHM onset directly [12]. Oxidative stress is also responsible for causing an inflammatory state that may also lead to the development of the disease [13]. In 2015, Zhu et al., and in 2021, Wei et al., determined that the microenvironment of EoHM eyes is unique due to the increase of proinflammatory cytokines (IL6, IFN-γ, IP-10, eotaxin, and MIP-1α) and angiogenic growth factors (VEGF, MCP1, and IL5) in the aqueous and vitreous humor found in patients [12,14,15,16]. The presence of a higher concentration of the vascular endothelial growth factor (VEGF) as an angiogenic factor could make the EoHM patients’ eyes more susceptible to developing vascular diseases [14].

This manuscript focuses on the genetic factors that may be affecting the development of EoHM, in which over 100 genes and 20 chromosomal loci have been identified in association with myopia and the related quantitative traits via linkage analysis, candidate gene analysis, genome-wide association study and next-generation sequencing (NGS). However, only a small number of genes have been identified to predispose individuals to EoHM, suggesting a complex mechanism [13,17,18]. Eleven genes have been identified as autosomal dominant genes to produce non-syndromic EoHM (*ZNF644*, *SCO2*, *SLC39A5*, *CCDC111*, *P4HA2*, *BSG*, *CPSF1*, *NDUFAF7*, *TNFRSF21*, *XYLT* and *DZIP1*); four as autosomal recessive (*LRPAP1*, *CTSH*, *LEPREL1* and *LOXL3*) and two linked to chromosome X (*ARR3* and *OPN1LW*). To date, 26 loci have been discovered (23 in autosomal chromosomes and 3 in chromosome X) [7,19]. Other studies have linked EoHM with genes such as *CTNND2*, *JOANA*, *CACNA1F* and *RPGR* [20,21]. Given that myopia is a refractive error disease, other studies have reported single nucleotide polymorphisms (SNPs) related to refractive error in *PRSS56*, *BMP3*, *KCNQ5*, *LAMA2*, *TOX*, *TJP2*, *RDH5*, *ZIC2*, *RASGRF1*, *GJD2*, *RBFOX1*, *SHISA6*, *FAM150B-ACP1*, *LINC00340*, *FBN1*, *DIS3L-MAP2K1*, *ARID2-SNAT1* and *SLC14A2* [22,23,24].

The proper diagnosis of EoHM is essential because it can be the first sign of a syndromic condition. For cases in which an NGS panel is performed with only non-syndromic EoHM-related genes, a syndromic condition, such as Stickler syndrome, might be overlooked. We therefore recommend a broader approach when faced with a child with EoHM. Several ophthalmological diseases, such as childhood glaucoma, retinopathy of prematurity, congenital stationary night blindness and cone-rod dystrophies, are known to be involved in the development of EoHM [25]. It is also essential to consider the family history in the early diagnosis, and treatment can prevent further complications.

This study is based on the hypothesis of the possible greater involvement of a genetic cause in the development of EoHM when compared with late-onset HM (LoHM). To clarify this, we analyzed 43 families with EoHM to describe not only the pathogenic genes that have already been related to EoHM or related syndromes but also new affected genes that have not been directly related and could be causing this condition.

## 2. Results

The study results were obtained from an ongoing project for identifying mutations that cause EoHM in a sample of families at a tertiary hospital in Spain. The project also aims to evaluate the implementation of NGS and its relevance as part of the approach for patients with EoHM.

A total of 43 patients with EOHM from 43 unrelated families (51% male [22/43] and 49% female [21/43]) were recruited based on their phenotype and inclusion criteria (listed in Section 4. Materials and Methods). A complete clinical evaluation and genetic analysis were performed for the proband of each of the 43 families. After performing the genetic analysis of all probands and their families, six genetically affected family members who had not been considered as probands were detected and were added to the study group.

The mean visual acuity on the decimal scale was 0.59 for the right eye (OD) and 0.55 in the left eye (OS). Axial length (AL) was measured with the IOL master 500 (Carl Zeiss Meditec, Jena, Germany) with a mean AL of 27.79 mm for the OD and 27.95 mm for the OS. Only 2 patients had an AL shorter than 25.5 mm. The mean spherical refraction was −10.8 SD for the OD and −10.44 SD for the OS, with a spherical equivalent of −11.22 for the OD and −10.44 for the OS (Table 1).

Strabismus was present in 12% of the patients (5/43), with esotropia being the most common type (80%). Twelve percent (5/43) of the patients presented with nystagmus.

When we consider only the 37 patients younger than 12 years at the time of the examination, only two (5%) presented with a posterior staphyloma, four (11%) presented with peripapillary atrophy, and 14 (38%) presented with diffuse chorioretinal atrophy. One patient had a colobomatous papilla, and one patient had complete retrolental retinal detachment in one eye (Table 2). We performed retinography with Optos (Marlborough, MA, USA) and optical coherence tomography (OCT) with Heidelberg OCT (Heidelberg Engineering GmbH, Heidelberg, Germany) (Figure 1, Figure 2, Figure 3 and Figure 4).

After performing an NGS analysis with a specifically designed panel, we found 46 variants in 33 probands in known genes for EoHM (Table 3). Of these, we found six pathogenic and five likely pathogenic variants (23.9%) and 35 (76.1%) variants of unknown significance (VUS) (Table 3). However, we found no genetic cause in 10 patients (23%).

Of the 11 variants classified as pathogenic and likely pathogenic, two families had an affected variant in *COL2A1* and another two had an affected variant in *COL11A1*, while the rest had an affected variant in *TRPM1*, *CACNA1F, PAX6, OPA1, ARL6, SCO2* and *NDP* (Figure 5). The rest of the variants were classified as VUS and were found in *TIMP2*, *COL9A1*, *CEP290*, *PCDH15*, *VDR*, *KCNV2*, *CFH*, *CACNA1F*, *COL2A1*, *ZNF644*, *CRYBB3*, *COL11A1*, *CRYGC*, *RDH5*, *IGF1R*, *MMP10*, *EPHA2*, *MERTK*, *CRYGA*, *ZNRF3*, *LAMA2*, *SCO2*, *MMP1* and *CRYBA1*. A number of genes with variants classified as VUS were found in two families (e.g., *GPR143*, *LRP5*, *PEX1* and *MMP9*) and in three families (e.g., *TRPM1*) (Figure 6).

Considering zygosity, 40 of the variants found were heterozygous (87%), whereas only one variant was homozygous (2%) and five were hemizygous (11%).

## 3. Discussion

In this study, we performed a mutational analysis of Spanish patients diagnosed with EoHM implementing a customized NGS panel containing 419 genes related to ophthalmological disorders with a suspected genetic cause. The use of an NGS panel strategy rather than whole exome sequencing as a first approach for the genetic study is in response to the need for a less expensive method that can provide efficient analysis with a low rate of incidental findings.

### 3.1. Genes Related to Isolated EoHM

As previously mentioned, a few genes are related to non-syndromic EoHM, such as *SCO2* and *ZNF644*. Two patients (from families OFT-00533 and OFT-00554) in our cohort showed previously reported variants in gene *SCO2* (OMIM:604272) [36,37,38,39], which encodes a copper chaperone essential for the formation of cytochrome c oxidase (COX), the last enzyme in the respiratory electron transport chain in the mitochondria [36]. The exact mechanism of myopia development associated with COX deficiency specifically is currently unclear, but several studies, such as the 2013 study by Tran-Viet et al. [36,37,40,41], have detected various mutations in gene *SCO2* that have been considered pathogenic or likely pathogenic for high or extreme myopia (>30 diopters). In contrast, Piekutowska-Abramczuk et al. (2016) [42] indicated that heterozygous pathogenic variants in this gene are not associated with high-grade myopia in either humans or mice. Considering the contradictory published data, more studies on the involvement of *SCO2* in the development of EoHM are needed to determine whether the variants reported in the patients are actually the cause of the phenotype or just a risk factor whose further development is affected by the environment. In our cohort, we confirmed that the variants are just a risk factor and not the cause, given that the proband of family OFT-00554 carried the same heterozygous variants as their healthy parents.

Another gene related to non-syndromic autosomal dominant EoHM is *ZNF644* (OMIM:614159), a zinc finger transcription factor expressed in the retina and the retinal pigment epithelium. The protein’s biological function has not been identified. As a transcription factor, however, it might regulate genes involved in eye development, triggering a mutant protein able to modify the axial elongation of the eye globe and cause EoHM [41,43]. Numerous studies have reported variants in this gene in non-syndromic EoHM [4,37,40,43,44]. We found one previously unpublished variant classified as VUS according to the American College of Medical Genetics and Genomics (ACMG) guidelines. The proband of this family (OFT-00332) had inherited the variant from their mother, who apparently did not have a pathogenic ocular phenotype, which suggests that this gene might present incomplete penetrance.

### 3.2. Genes Related to Inherited Retinal Diseases

Inherited retinal dystrophies are sometimes associated with EoHM, which might be the first isolated sign at presentation. Of the 46 variants found in our cohort, ten were found in six genes associated with inherited retinal degeneration (based on the Retinal Information Network [RetNet]) (21.7%, 10/46): *TRPM1* (OMIM:603576), *CACNA1F* (OMIM:300110), *KCNV2* (OMIM:607604), *MERTK* (OMIM:604705), *RDH5* (OMIM:601617) and *ARL6* (OMIM:608845).

We reported 4 variants in *TRPM1*, a gene associated with congenital stationary night blindness (CSNB), a genetically and clinically heterogeneous disease that manifests as non-progressive nyctalopia and with an electronegative full-field electroretinogram. Our cohort had one patient (from family OFT-00074) with heterozygous compound variants in this gene who presented with nystagmus, esotropia and EoHM. The patient underwent an electroretinogram, which showed moderate-severe disease in the peripheral retina, mild-moderate disease in the central retina and an electronegative result for the bilateral scotopic response, which suggests a functional disorder of the bipolar cells. According to Miraldi Utz et al. (2018), patients presenting EoHM, strabismus, nystagmus and a variant in this gene should undergo full-field electroretinography, given that a considerable proportion of these patients might not present nyctalopia at the start and that the procedure could help diagnose CSNB [26]. In addition, two more VUS were reported, in families OFT-00453 and OFT-00490.

Another gene associated with the same phenotype of CSNB is *CACNA1F*, a gene present in the X chromosome and related to Aland Island disease and cone-rod dystrophy. Wutz et al. (2002) mentioned the possibility of hemizygous variants in this gene causing a phenotype of CSNB with EoHM as one of their most common features [21,45,46]. Our study had two families (OFT-00155 and OFT-00097) with hemizygous variants and a phenotype of EoHM and nystagmus. For family OFT-00155, the variants were maternally inherited, and the proband had a brother with the same phenotype who also presented this variant.

One patient from family OFT-00391 had a variant in *RDH5*, a gene expressed in the retinal pigmented epithelium and involved in the retinoid cycle, the metabolic pathway that regenerates the visual chromophore following light exposure [22,47]. The phenotype related to variants in this gene is fundus albipunctatus, a form of CSNB [48]. Although the patient did not have retinopathy, this gene was established in 2013 as a susceptibility gene for refractive error and myopia [18,19,20,21,22,23,24,25,26,27,28,29,30,31,32,33,34,35,36,37,38,39,40,41,42,43,44,45,46,47,48,49,50].

Patients from three families with a phenotype of EoHM and retinal dystrophy had variants in *KCNV2* (OFT-00092), *MERTK* (OFT-00474) and *ARL6* (OFT-00407), genes with a phenotype of retinal cone dystrophy in the case of *KCNV2* and retinitis pigmentosa in the other two cases. *KCNV2* has EoHM as one of its ocular features. In the case of *MERTK* and *ARL6* where EoHM was not reported, however, this manifestation might be a second consequence of the genetically determined retinal dystrophy.

### 3.3. Genes Related to Vitreoretinal Inherited Diseases

Regarding the association between EoHM and vitreoretinal inherited diseases, Marr et al. (2001), showed an even higher association between HM and systemic and other ophthalmological conditions, taking into account that they did not consider an exclusion criterion the presence of a syndromic phenotype as an exclusion criteria, as we did in our study. Out Of the 112 probands, the authors found that only 8% had non-syndromic HM, 54% had an underlying systemic association, and the remaining 38% remaining had further HM-related ocular problems associated with HM [17]. Other studies, such as that performed by Logan et al. (2004) analyzed the genetic results of children with EoHM diagnosed before 10 years of age. Fifty-six percent of the children presented with simple HM, 25% were diagnosed with inherited retinal dystrophies and amblyopia, and 19% were diagnosed with an HM-related systemic disorder [51].

Based on these results, the RetNet genes associated with vitreoretinal inherited diseases with syndromic manifestations should be included for patients with EoHM, such as *COL9A1* (OMIM:120210), *COL2A1* (OMIM:120140), *COL11A1* (OMIM:120280), *CEP290* (OMIM:610142), *PCDH15* (OMIM:605514), *LRP5* (OMIM:603506), *PEX1* (OMIM:602136), *CFH* (OMIM:134370), *OPA1* (OMIM:605290) and *NDP* (OMIM:300658).

Fifteen percent of the reported variants in our study were in three collagen genes (*COL2A*, *COL9A1* and *COL11A1*), divided into four pathogenic and likely pathogenic variants and three VUS in OFT-00181(whose variant was inherited from the affected mother), OFT-00191 (from the affected father), OFT-00209, OFT-00275, OFT-00453, OFT-00490 and OFT-00590.

These genes are included in the collagen superfamily of proteins that have an essential role in the structural and mechanical properties of tissues and, more specifically, in the connective tissue [52,53]. If we consider only the variants found in *COL2A1* and *COL11A1*, the percentage of mutations in these two genes in our cohort was 13%, which is slightly higher than the 5% reported by Sun et al. in 2015 [54]. This result can be explained by the size of the authors’ cohort (298 probands), which can cause the percentage of collagen genes to decrease when variants appear in other genes. The authors also used whole exome sequencing analysis, which can result in more variants in different genes.

These three genes are considered to cause Stickler’s syndrome, a connective tissue disorder caused by mutations in collagen genes, which can be inherited as an autosomal dominant disorder if *COL2A1, COL11A1* or *COL11A2* (OMIM:120290) are affected, or as autosomal recessive if *COL9A1*, *COL9A2* (OMIM:120260) and *COL9A3* (OMIM:120270) are affected [55]. This syndrome can manifest as a systemic disease or as a mainly ocular condition. Despite the variability in phenotypic expression that can occur within and among families, the basic features of the disease include abnormal vitreous findings (which is a pathognomonic feature), high myopia in more than 90% of patients [54], orofacial abnormalities, arthropathy and varying degrees of deafness. Stickler’s syndrome is the most common cause of inherited rhegmatogenous retinal detachment, which can lead to severe vision loss.

Our study patients were recruited based on the fixed inclusion criteria; therefore, the initial diagnosis was of EoHM, and the finding of the genetic alteration enabled the diagnosis of a possible syndromic disease that had not been previously suspected. A funduscopic examination under sedation was performed on the patients with mutations in one of these three collagen genes, where pathognomonic vitreous alterations for Stickler’s syndrome were found, leading to the assessment of the patients by other specialists. This was the case in our cohort with families OFT-00181 and OFT-00275, in which the genetic diagnosis allowed for a preventive treatment for the existing retinal lesions that could have led to further complications.

The importance of an early diagnosis of Stickler’s syndrome relies on closer follow-up by various specialists to evaluate other possible phenotypic disorders, such as facial alterations, cleft palate, elbow hypermobility, femoral head necrosis and valgus knee. Specifically, it allows for a more thorough evaluation by the ophthalmologist to anticipate possible complications and perform preventive treatment to prevent vision loss in these patients. Considering Stickler’s syndrome in patients with HM before 10 years of age is therefore highly relevant, even if it appears to be the only manifestation, given that it could be the presenting symptom [55,56].

Other syndromes with EoHM included in the ocular phenotype are Marfan syndrome and homocystinuria [57,58]. There are other cases in which EoHM is the secondary feature, as is the case of the proband patient from family OFT-00559 who had a phenotype of EoHM and nystagmus and a pathogenic variant in gene *NDP* that encodes the norrin protein, a secretory growth factor that regulates retinal angiogenesis [59]. The reported phenotypes related to this gene are Norrie disease, Coats disease and X-linked exudative vitreoretinopathy. The overlap between the clinical features complicates the diagnosis. It is therefore important to determine the age at onset, hearing difficulties, central nervous system abnormalities and the family history when performing diagnosis and genetic counseling [60]. In this case, the family history helped with the diagnosis, given that the proband had two cousins already diagnosed with Norrie disease. The examination under anesthesia after the genetic results showed avascular areas of the retina that required laser treatment. Knowledge of the presence of this mutation enables a more thorough retinal examination and preventive treatment that, having been missed, could have resulted in retinal detachment or further visual loss. In the case of this family, EoHM was a consequence of a genetically determined retinal disease.

In this study, one patient from family OFT-00177 with EoHM and retinal dystrophy had two VUS, one in *CEP290* (a gene related to Bardet-Biedl syndrome, Meckel syndrome, Joubert syndrome and Senior-Løken syndrome) and the other in *PCDH15* (a gene responsible for Usher syndrome). Despite the fact that these two genes do not have a phenotype of high myopia, Wan et al. (2018) reported *CEP290* and *PCDH15* as novel candidate genes of myopia pathogenesis [61]. Young et al. (2006) reported that *PCDH15* is included in a novel locus for high-grade myopia [60].

Families OFT-00178 and OFT-00332 had variants in gene *LRP5* that are related to syndromes without ocular features and exudative vitreoretinopathy. Chen et al. (2018) observed the presence of a synonymous variant in this gene as pathological and the cause of EoHM presented by their patient. The authors suggested that although the variant does not alter the amino acid chain, the altered nucleotide can modify the secondary structure of the protein and can influence protein holding and gene expression [50]. *PEX1* is a gene whose variants are related to defects in peroxisome biogenesis. In our cohort, two families (OFT-00223 and OFT-00568) had VUS in this gene. Poll-The et al. (2004) indicated that variants in this gene can produce numerous phenotypes, EoHM being one common feature present in their patients [4,62].

Another gene of interest is *CFH*. Basal laminar drusen is the only ocular phenotype reported with a mutation in *CFH*, but variants can also produce complement factor H deficiency or hemolytic uremic syndrome. In the last few years, various researchers have tried to link EoHM to changes in *CFH*. In 2021, García-Gen et al. established the key role of *CFH* in the development of myopic disease [63]. In our study, there was a variant in this gene in OFT-00097.

Previous studies have proven the relationship between alterations in *OPA1* and high myopia [17,64]. However, *OPA1* is known for its relevance in autosomal dominant optic atrophy (ADOA), which is the most common form of hereditary optic neuropathy. In fact, *OPA1* mutations are responsible for approximately 90% of cases [64]. The absence of a normal *OPA1* protein results in abnormal mitochondrial cristae, the release of cytochrome c and apoptosis [65]. In our cohort, 1 family (OFT-00343) had a likely pathogenic mutation in this gene, with two sisters affected and no reported phenotype for the parents. The variant was inherited from the father. Variable expressivity and incomplete penetrance were found in this gene, which could be why the variant was also found in the father, who did not have a reported phenotype [66]. Chen S et al. (2007) observed a family with ADOA, a novel *OPA1* mutation, and a higher frequency of EoHM [50]. Nevertheless, the presence of EoHM in one family member without ADOA and *OPA1* challenged the assumption that this mutation induced EoHM. ADOA is known to induce a thinning of the retinal nerve fiber layer and of the ganglion cell layer, inducing a reduction in total macular thickness [67]. Li C et al. (2005) observed the same results in a family with ADOA and hearing loss who presented with myopia; however, it was not possible to demonstrate that EoHM was a direct consequence of the mutation of this gene [65].

### 3.4. Other Genes

In contrast to the above, there were a few genes reported in our study that are not clearly considered to cause EoHM or retinal dystrophy but could be related to the onset of EoHM. Given that EoHM is characterized by scleral thinning, we studied another family of proteins: matrix metalloproteinases (MMPs). In this case, we found VUS in the three genes responsible for MMPs: *MMP1* (OMIM:120353), *MMP9* (OMIM:120361) and *MMP10* (OMIM:185260) in families OFT-00223, OFT-00429, OFT-00436 and OFT-00586. In the case of family OFT-00436, the variant was inherited from the affected mother. The relevance of studying this family of proteins relies on their involvement in organizing the tissue, given that the proteins are responsible for the degradation of collagen and other extracellular matrix components. The structural organization and constant remodeling of the sclera is highly dependent on the activity of the fibroblast, which is the major extracellular matrix-producing cell [68]. Among these proteins, MMP1, MMP2 (OMIM:120360), MMP3 (OMIM:185250), MMP9 (gelatinase B) and MMP14 (OMIM:600754) have been shown to be expressed in human sclera. These MMPs are therefore potentially responsible for scleral remodeling [59]. MMP2, the most studied sclera metalloproteinase, causes scleral collagen degradation when it increases its activity [69].

Given that the increase in myopia in highly myopic patients is almost always associated with an increase in AL, a thinning of the sclera occurs, mainly at the posterior pole. The most relevant consequence is the potential formation of a posterior staphyloma, an area in which the thin sclera becomes ectatic. Even without staphyloma, when the sclera of a highly myopic patient is compared with that of a non-myopic patient, the highly myopic sclera is up to 50% thinner [68].

Due to this constant remodeling and thinning of the sclera, changes in the myopic fundus increase as patients age [70]. When considering only those patients in our study younger than 12 years at the time of examination, only 5% (2/37) presented with a posterior staphyloma, 11% (4/37) presented with peripapillary atrophy and 38% (14/37) presented with diffuse chorioretinal atrophy. These findings are significantly different from those of studies in older patients, such as that by Gozum et al., which showed posterior staphylomas in 23.6% of cases and peripapillary crescents in 66.5% [71], results that agree with those of Jagadeesh et al. (2020), who observed a higher prevalence of myopic funduscopic changes in older patients [72]. These findings are consistent with the constant corneal remodeling and thinning performed by these MMPs.

The variation in these *MMP* genes’ expression in the sclera can lead to greater susceptibility to increased axial elongation of the eye [73]. According to Yue et al. (2020), this increase in AL is related to an excessive degradation and reduced synthesis of the scleral extracellular matrix [74]. The authors found a positive correlation between MMP2 levels in the aqueous humor and AL, which supports the hypothesis that misregulation of this protein might be responsible for a higher degree of scleral remodeling and therefore increased AL. However, the authors found no correlation between plasma MMP levels and AL, supporting the hypothesis of a more local alteration of MMPs in the sclera.

The models used by David et al. (1997) indicated that the mechanical stress on the retina and choroid during eye movements is significantly higher in larger eyes than smaller eyes [75]. According to an in vitro study by Shelton et al. (2006), mechanical strain stimulates *MMP2* activation by scleral fibroblasts, which contributes to scleral extracellular matrix degeneration, scleral thinning and possible ocular ectasia [76,77].

Other studies, however, such as that by Schache et al. (2012) with an Australian cohort and the one by Nakanishi et al. (2010) with a Japanese cohort, found no association between mutations in *MMP1*, *MMP2*, *MMP3*, *MMP8* (OMIM:120355), *MMP9*, *MMP10*, *MMP11* (OMIM:185261), *MMP13* and EOHM [73,78].

The development of myopia has been associated with a mean reduction in *TIMP2* (OMIM:188825) mRNA expression, a variant of which was present in family OFT-00209. The activity of *MMP2* and *TIMP2* is correlated to increased *MMP2* activity over a critical point, inducing inhibition of *TIMP2* activity, thereby favoring collagen degradation [69]. A study by Leung et al. (2011) analyzed data from an adult Han Chinese population and observed that *MMP2*, *TIMP2* and *TIMP3* (OMIM:188826) genes were not associated with high myopia in their cohort [69]. However, in a study by Zhuang et al. (2014) measuring MMP2, TIMP2 and TGFB2 (OMIM:190220) levels in vitreous samples from patients who underwent vitrectomy, the MMP2/TIMP2 ratio was significantly higher in the vitreous samples from the EoHM group than from those of the control group. *MMP* activity was also significantly higher in the vitreous samples from the EoHM group than in those of the control group [79,80]. From these studies, we can deduce that the elevated MMP/TIMP ratio and *MMP* activity could play a role in the pathogenesis of human high myopia. However, given that studies that have analyzed the relevance of *MMP2*, *TIMP2* and *TIMP3* in blood samples have shown no differences between patients with EoHM and controls, further studies are needed to evaluate whether the importance of *MMPs* and *TIMPs* relies on changes on a more local level.

Patients with EoHM appear to have more lens changes [81,82]. The lens is a transparent ellipsoid organ located in the anterior segment of the eye. Due to the refractive medium of the core, the lens is responsible for the full range of vision [82]. Although the mechanism of myopia progression has been published [83,84], lens changes in highly myopic eyes and their molecular pathogenesis are still unknown [85,86]. In recent years, numerous studies have been conducted to demonstrate that lens overgrowth is related to myopia [87], resulting in a new hypothesis: the lens diameter of a highly myopic eye might be larger than that of an emmetropic eye [87]. For the increased lens diameter, authors have suggested the continuous production and accumulation of structural proteins, mainly crystallins (which include three families: α, β and γ), given that they account for 90% of these proteins. Previous studies have reported a decrease in the soluble expression of α-crystallins in lenses of patients with high myopia [88,89]. Zhu et al. (2021) [82] suggested that changes in β and γ-crystallin should also be related to the development of EoHM. The authors reported that the increase in these proteins might be due to an increase in the expression of genes in these two families but might also be produced by the activation of MAF transcription factor that activates crystallin proteins downstream. Considering the potential relationship between changes in β and γ-crystallin expression and EoHM, we observed four VUS in four genes (*CRYBB3* (OMIM:123630), *CRYGC* (OMIM:123680), *CRYGA* (OMIM:123660) and *CRYBA1* (OMIM:123610) in four families OFT-00332, OFT-00391, OFT-00493 and OFT-00630, which have not been previously published.

Another of the EoHM-related genes is *GPR143* (OMIM:300808), which is expressed in the retinal pigment epithelium and is associated with the development of albinism and nystagmus [90]. We found two VUS in these genes: one in a patient with EoHM (OFT-00601) and the other in a patient with EoHM and nystagmus (OFT-00155). In these two probands, the variant was inherited from the mother, who was also affected.

We found one variant in the *VDR* (OMIM:601769) gene in OFT-00223, which had already been reported as a phenotype of vitamin D resistance but is considered a risk factor for developing EoHM [91,92,93].

*LAMA2* (OMIM:156225) is considered to produce muscular dystrophy with very few ocular features. Laminins are structural proteins that are the main components of the extracellular matrix, whose changes in its composition in the sclera are related to an AL alteration. Specifically, laminin alpha 2 is expressed in the sclera and optic nerve, is present in extraocular muscles during development and can act as a guide for retinal ganglion cell growth. Kiefer et al. (2013) and Cheng et al. (2013) reported that this gene can be considered one of the multiple factors that act in the development of myopia [22,94,95]. In our study, family OFT-00546 had a variant in *LAMA2*. Another gene related to AL abnormalities is *ZNRF3* (OMIM: 612062) [69], a protein involved in the Wnt signaling pathway; family OFT-00506 had a variant in this gene.

A patient from family OFT-00463 had a variant in *EPHA2* (OMIM:176946), a gene related to age-related cortical cataracts. This phenotype was reported to have EoHM as an ocular feature [4,96]. Considering that the patient was 57 years old, this gene could be responsible for the EoHM, with no signs of cataract at present.

A patient from family OFT-00045 had a variant in *PAX6* (OMIM:607108), the paired box 6, which is considered a master gene for eye development. The gene’s best-known phenotype is aniridia. According to a meta-analysis by Tang et al. (2014 and 2018), there is a suggestive association between *PAX6* and extreme/high myopia but not lower grade myopia, although further studies are necessary for validation [97,98]. *PAX6* plays an important role in controlling eye globe growth, according to Bilbao-Malavé et al. (2020), and has been shown to have a suggestive association with EoHM, as demonstrated in a meta-analysis that included studies mainly performed on Asian populations [99,100,101].

Lastly, a patient from family OFT-00429 presented a mutation in *IGF1R*, which encodes for IGF1R, a transmembrane receptor tyrosine kinase responsible for mediating proliferation and protection from apoptosis. A study by Wang et al. observed no relationship with mutations in *IGF1R* or *IGF1* in the development of EoHM in a Chinese population [102]. However, further studies are necessary to rule out *IGFR1* mutations as a cause of EoHM, given that a study performed on chickens by Penha et al. (2011) showed that overminus lens therapy, and therefore hyperopic defocus, influenced the expression of insulin receptor and *IGF1R* receptor in the retinal pigment epithelium. The shift of the focal plane behind the photoreceptor layer triggers substantially increased eye growth [102,103].

As mentioned above, high myopia may be related to the microenvironment as well as genetics. In 2017, Fedor et al. demonstrated that serum zinc and copper levels in patients with EoHM were lower than those of the control group. As these two factors are antioxidant elements, the existence of an oxidant/antioxidant imbalance in patients with EoHM is suggested [104]. In 2020, Mérida et al., compared antioxidant-oxidant status in aqueous humor samples of myopic and non myopic patients. They measured the total antioxidant capacity (TAC) as an indicator of the overall capacity to neutralize ROS and the total concentration of nitrite/nitrates. TAC levels were lower in the highly myopic group which indicates that the level of ROS had increased and that the aqueous humor of these patients was undergoing oxidative stress. They also demonstrated that the lower TAC was directly related with the grade of myopia, which suggests that knowing the concentration of these factors could help preventing and treating these myopic patients before severe complications appear [105].

Mérida et al. also found an increase in the total nitrite levels, the end product of nitric oxide metabolite, in highly myopic patients. This parameter is considered as one of the origins of oxidative stress damage and reduced antioxidant capacity, but nowadays it has been suggested to play a role as a nitric oxide reservoir that produces nitric oxide in hypoxia conditions. Finally, it is known to act as an eye modulator in developing myopia, acting in ocular growth, intraocular pressure regulation and retinal vascular development. Despite this, it is not easy to understand the results because of the neurodegenerative/neuroprotective role of the nitric oxide. In our panel, we found variants in five genes that have been linked to oxidative stress (*ZNF644*, *TIMP2*, *MMP1*, *MMP9* and *MMP10*). Another four genes of the panel, *HGF*, *RBP3*, *MMP2* and *MMP3,* have been related to this factor even though the patients did not show any variant in them [105,106,107,108]. Further studies are needed to clarify the effect of these parameters, but oxidative stress may help explain the development of myopia in some patients [105] as it is associated with other eye diseases such as cataract, glaucoma, age-related macular degeneration and dry eye syndrome [108,109].

EoHM may be associated with more severe ocular complications such as choroidal neovascularization (CNV) affecting 5–10% of these patients. Pathological myopia is the main cause of onset of CNV in patients under 50 years of age [110]. The presence of a higher concentration of vascular endothelial growth factor (VEGF) as an angiogenic factor could make the patients’ eyes more likely to develop vascular diseases. This finding may be the explanation of the increase of CNV cases in patients with EoHM. In addition, three genes of our panel were related to vascular diseases such as *ARSM2*, *HTRA1* and *CFH*, in which a variant was found in family OFT-00097 [111]. In those patients where an aqueous or vitreous humor sample is available, it would be interesting, in future studies, to measure the VEGF concentration to study this correlation [15].

As previously mentioned, the hypothesis of our study is based on is the potential greater involvement of genetics in the development of EoHM than of LoHM. Zhou et al. (2018) compared the mutation detection rate in a study of LoHM with one of their previous studies, considering the possible involvement of RetNet genes. The authors found a genetic cause in only 7.2% of the patients with LoHM and in 23.8% of those with EoHM [3,13], a result similar to the 21% (9/43) of our study patients with pathogenic and likely pathogenic variants in RetNet genes. This provides further genetic evidence that EoHM differs genetically from LoHM and that genetic testing of patients with EoHM is necessary to find the cause of the disease.

## 4. Materials and Methods

The combined ophthalmological and genetic approach was performed by the Ophthalmogenetics Multidisciplinary Unit at La Paz University Hospital, Madrid, Spain, according to the tenets of the Declaration of Helsinki and was approved by the ethics committee.

***Participants:*** A total of 49 patients from 43 unrelated families aged 4–74 years were recruited for the study, as well as both parents of the proband for those cases in which it was possible. After informing and obtaining written consent from the proband and from the patients’ parents or their guardians, a peripheral venous blood sample was collected to isolate the genomic DNA from leukocytes.

The inclusion criteria were as follows: (1) bilateral myopia with a refraction error ≥−6 diopters in at least one eye with onset before 10 years of age; (2) absence of cataracts; (3) absence of corneal disease or other ophthalmological diseases that produce secondary high myopia; and (4) absence of syndromic phenotype. Patients were recruited according to phenotype and inclusion/exclusion criteria and then underwent a genetic analysis.

***Clinical evaluation:*** A complete ophthalmological evaluation of the proband was performed, which included best corrected visual acuity, refraction before and after cycloplegia, funduscopic examination, AL measurement, retinography images and OCT.

***Genetic analysis:*** Genomic DNA was isolated from leukocytes in peripheral venous blood samples in the preanalytical area of our institute using the commercial Chemagic Magnetic Separation Module I (Chemagen, PerkinElmer, Waltham, MA, USA). DNA concentrations were measured by spectrofluorometer quantification using a NanoDrop ND-1000 Spectrophotometer (ThermoFisher Scientific, Waltham, MA, USA). Paired-end libraries were created using 1 µg of genomic DNA with KAPA HyperPrep Kit (Roche, NimbleGen, Inc., Pleasanton, CA, USA) and hybridization with a KAPA HyperCapture Reagent Kit (Roche, NimbleGen, Inc., USA).

***Custom panel design:*** The panel was captured using OFT-v3-1 design (Appendix A) and sequencing was done on the Illumina HiSeq 4000 platform (Illumina, Inc., San Diego, CA, USA). The data produced were aligned and mapped to the human genome reference sequence (GRCh37/hg19). The strategy for screening mutations was based on the use of NGS, implementing a customized panel (OFTv3.1) including 419 genes related to ophthalmological disorders with a suspected genetic cause (Appendix A), including 93 genes and regions related to EoHM pathogenesis or within EoHM-related loci: *ACAN, ACTC1, ADORA2A, AP3B2, BLID, C3orf26, CD55, CFH, CHRM1, CHRM2, CHRM3, CHRM4, CHRM5, COL18A1, COL1A1, COL2A1, CRISP3, CRYAB, CTNND2, CYP17A1, CYP19A1, EGR1, FGF10, FGF2, FHIT, FMOD, FOS, FSCB, GC, GJD2, HGF, HSD17B1, HSD3B1, IGF1, IGF1R, IGF2, IGF2R, IGFBP1, IGFBP3, IGFBP4, INS, INSR, IRS1, JUN, KERA, LAMA1, LAMA2, LUM, LYPLAL1, MET, MIPEP, MMP1, MMP10, MMP2, MMP3, MMP9, NAP1L4, PML, PPFIA2, PROM1, PTCHD2, PTPRR, RAD21, RARB, RASGRF1, RDH8, RSPO1, SCO2, SHISA6, SLC30A10, SNTB1, SRD5A2, TCF4, TCTE1, TGFB1, TGFB2, TGIF1, TIMP1, TIMP2, TIMP3, TOX, UHRF1BP1L, UMODL1, VCAN, VDR, VIP, VIPR2, WNT7B, ZEB2, ZIC2, ZNF644, ZNRF3, ZWINT*; chr 1: 219782981-219785408, chr 7: 130561506-130561569, chr 8: 8905955-8906028, chr 8: 9760898-9760982 and chr 10: 59064239-59064319.

The OFTv3.1 panel was designed with NimbleDesign software Roche NimbleGen, Inc., Pleasanton, CA, USA): HG19 NCBI Build 37.1/GRCh37, the target bases covered 99.5% and the size was 1,245,179 Kb. The mean horizontal coverage was 99.91%, and the mean sequencing depth per nucleotide was 467.

***Bioinformatic analysis:*** The first analysis was performed by the Institute of Medical and Molecular Genetics (INGEMM) Clinical Bioinformatics team, who developed an analytical algorithm that identifies point polymorphisms (SNP) and insertions and deletions of small DNA fragments inside the capture regions that are included in the NGS panels. The system comprises a sample pre-processing step, alignment of reads to a reference genome, identification and functional annotation of variants, and variant filtering. All these steps employ open tools widely used in the scientific community, as well as proprietary tools. Furthermore, all phases are designed in a robust manner and include statistical parameters that provide information on the status of the process and the convenience of continuing with the analysis. This system allows for the monitoring of the process and the appropriate quality controls to issue a reliable report on the aforementioned variants. Lastly, the system backs up the raw and processed data, which are stored in a database using encrypted and anonymized records to preserve patient confidentiality.

The bioinformatics analysis was performed by the Clinical Bioinformatics Unit of the INGEMM center using the following software tools: trimmomatic-0.36, bowtie2-align version 2.0.6, picard-tools 1.141, samtools version 1.3.1, bedtools v2.26 and GenomeAnalysisTK version 3.3-0. The databases employed were dbNSFP version 3.5, dbSNP v151, ClinVar date 20180930, ExAC-1, SIFT ensembl 66, Polyphen-2 v2.2.2, MutationAssessor, release 3, FATHMM, v2.3, CADD, v1.4 and dbscSNV1.1.

***Genotype-phenotype correlation:*** The second analysis consisted of evaluating the pathogenic clinical significance of the variants found in the patients by employing multiple databases. The first and most extended database is VarSome, which contains an algorithm that uses the 2015 ACMG guidelines [112], based on a combination of previous reports in the literature and computational, functional and population data as reference, providing the classification of the variants as pathogenic, likely pathogenic, VUS, likely benign or benign. Those variants in the proband and other family members classified as pathogenic, likely pathogenic or VUS according to ACMG guidelines were validated using Sanger sequencing or other techniques, if possible.

***Family studies:*** When the variant was fully validated in the proband and the family members, the third part of the analysis consisted of studying the phenotypes related to the gene, to determine whether the variant being studying was the causative variant (Figure 7). To address this, we used the online catalog of human genes and genetic disorders known as the Online Mendelian Inheritance in Man [113], the available literature and specific databases such as RetNet. To check whether the variant had been previously reported, we used the Human Gene Mutation Database Professional.

## 5. Conclusions

Eleven families in our study had pathogenic and likely pathogenic variants, 1one family had a variant in *SCO2*, a gene related to non-syndromic EoHM, three had variants in genes associated with other retinal dystrophies (*TRPM1*, *CACNA1F* and *ARL6*), six had variants in genes related to a syndromic disease that feature retinal dystrophy (*COL2A1*, *COL11A1*, *OPA1* and *NDP*) and one had a variant in *PAX6* that is not included in any of the groups mentioned but was reported to have a suggestive association with EoHM development.

Furthermore, 24 families had VUS, two families with variants in genes related to non-syndromic EoHM (*SCO2* and *ZNF644*) and seven families with variants in genes associated with other retinal dystrophies (*TRPM1*, *CACNA1F*, *KCNV2*, *RDH5* and *MERTK*). These results support the hypothesis previously indicated by other researchers who found 25% and 38% of variants related to this type of ocular disease and who indicated that EoHM could be a secondary consequence of a genetically determined retinal dystrophy. Ten families had variants in genes related to a syndromic disease with retinal dystrophy (*COL9A1*, *CEP290*, *PCDH15*, *LRP5*, *PEX1*, *CFH*, *COL2A1* and *COL11A1*). Genetic analysis of patients with EoHM is relevant to anticipate possible syndromic manifestations and other ophthalmological conditions, especially in cases when the disease is present as a non-typical phenotype. The detection of variants that indicate other than a simple EoHM allow these patients to be managed by multidisciplinary units and thereby perform more global patient care, preventive treatment in those cases where it is possible and providing closer follow-up to prevent vision loss. Lastly, 14 families had at least one variant in genes such as *GPR143*, *TIMP2*, *VDR*, *MMP1*, *MMP9*, *MMP10*, *PAX6*, *CRYBB3*, *CRYGC*, *CRYGA*, *GRYBA1*, *IGF1R*, *EPHA2*, *ZNRF3* and *LAMA2*. Although these genes are not clearly considered to cause EoHM or retinal dystrophy, the published information could relate them to the onset of EoHM. Further studies are needed to confirm the hypothesis of the involvement of these genes.

The genetic alterations for most patients with EoHM have not yet been defined, with the causative variant found in only a small number of patients. The genetic study of this disease is currently focused on evaluating every single gene suspected of being a candidate of EoHM. We found no candidate genes in 23% of the patients, which suggests that whole exome sequencing and whole genome sequencing studies, animal models and other studies are needed to clarify the disease’s genetic cause.

Based on the above, we can state that EoHM is a complex disease based on a combination of genetic, environmental and microenvironmental components. We recommend a systematic genetic analysis of patients with EoHM and of their relatives, given that it promotes better management of family members, helping with follow-up, prognosis and genetic counseling.

## Figures and Tables

**Figure 1 ijms-23-04233-f001:**
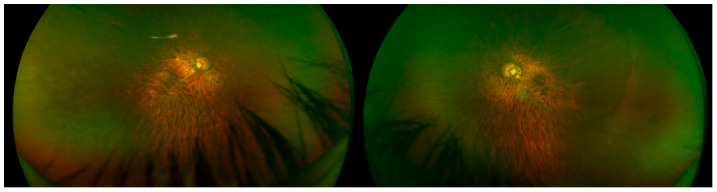
Posterior staphyloma. Right and left eyes. Optos (Marlborough, MA, USA).

**Figure 2 ijms-23-04233-f002:**
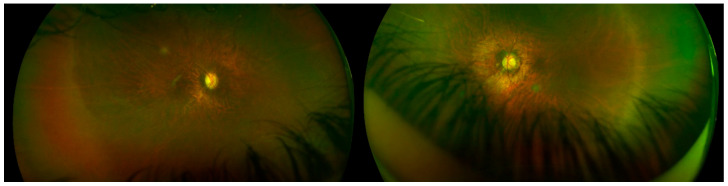
Diffuse chorioretinal atrophy. Right and left eyes. Optos (Marlborough, MA, USA).

**Figure 3 ijms-23-04233-f003:**
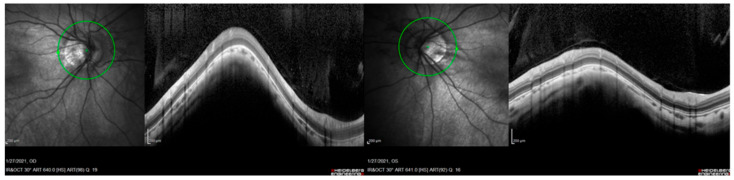
Optic nerve OCT with peripapillary atrophy, from left to right, OD and OS. Patient from family OFT-0047.

**Figure 4 ijms-23-04233-f004:**
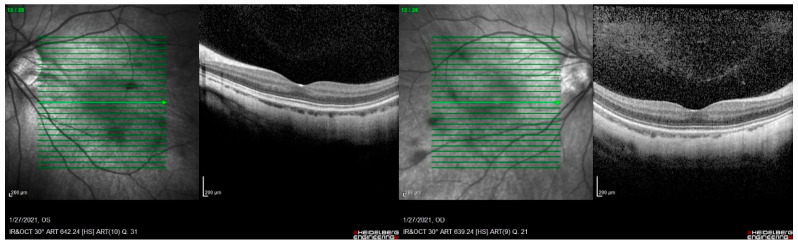
Macular OCT with central staphyloma, from left to right OD and OS. Patient from family OFT-0047.

**Figure 5 ijms-23-04233-f005:**
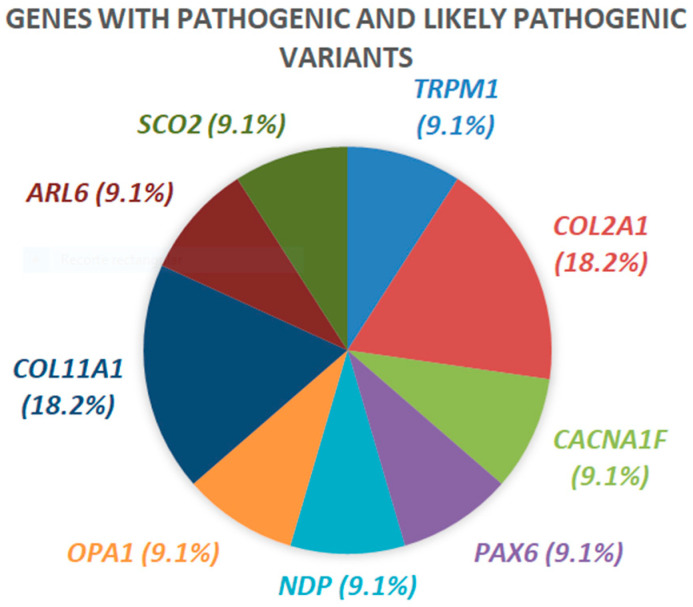
Representation of genes with pathogenic and likely pathogenic variants and their proportion.

**Figure 6 ijms-23-04233-f006:**
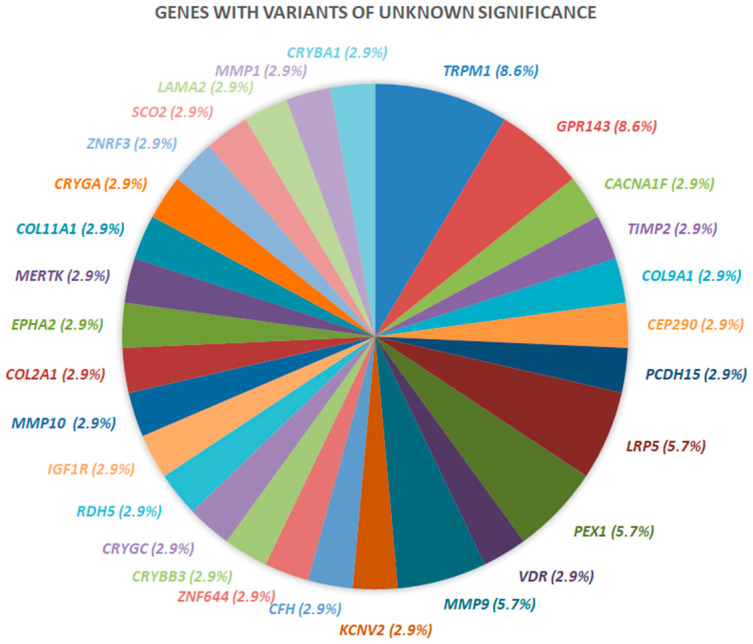
Representation of genes with variants of unknown significance and their proportion.

**Figure 7 ijms-23-04233-f007:**
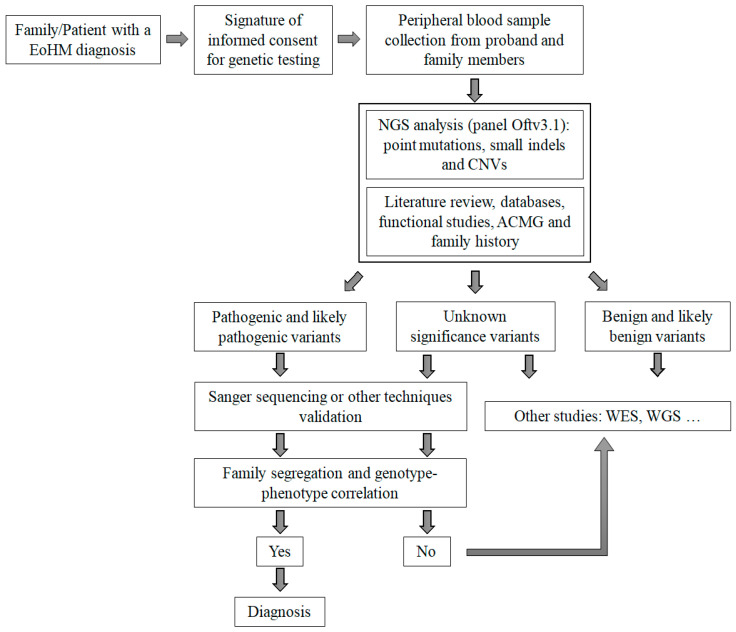
Diagnostic algorithm for early-onset high myopia. EoHM, early-onset high myopia; CNVs, copy number variants; ACMG, American College of Medical Genetics; NGS, next-generation sequencing; WES, whole exome sequencing; WGS, whole genome sequencing.

**Table 1 ijms-23-04233-t001:** Refractive results.

	Right Eye	Left Eye
Best corrected visual acuity (decimal scale)	0.59 ± 0.33	0.55 ± 0.33
Axial length (mm)	27.79 ± 2.5	27.95 ± 2.59
Spherical refraction (diopters)	−10.8 ± 6.1	−10.44 ± 5.38
Astigmatism (diopters)	−1.71 ± 1.3	−1.92 ± 1.4
Spherical equivalent (diopters)	−11.22 ± 5.45	−10.44 ± 4.66

Results are presented as mean ± standard deviation.

**Table 2 ijms-23-04233-t002:** Clinical Evaluation of Probands.

Family Number	Sex	BCVA OD	BCVA OS	AL OD	AL OS	Funduscopic Examination OD	Funduscopic Examination OS	SPcc OD	Astig OD	SE OD cc	SPcc OS	Astig OS	SE OS cc
OFT-00074	F	0.6	0.08	26.6	26.93	Diffuse chorioretinal atrophy, Central staphyloma	Diffuse chorioretinal atrophy, Central staphyloma	−12	−0.5	−12.25	−12.75	−2.5	−14
OFT-00155	M	0.125	0.1	NA	NA	Healthy retina	Healthy retina	−10	−1.25	−10.6	−8.75	−2.75	−10.1
OFT-00209	M	0.6	0.7	NA	NA	Diffuse chorioretinal atrophy	Diffuse chorioretinal atrophy	−8.5	−3	−10	−7	−3	−8.5
OFT-00177	F	NA	NA	NA	NA	Diffuse chorioretinal atrophy	Diffuse chorioretinal atrophy	−24	0	−24	−18	0	−18
OFT-00178	M	0.3	0.4	26.75	26.65	Healthy retina, Mild optic nerve pallor	Healthy retina	−6.75	−4	−8.75	−7.25	−3.25	−8.88
OFT-00181	M	0.9	0.9	26.6	26.7	Healthy retina	Healthy retina	−8.25	−1	−8.75	−8	−1	−8.5
OFT-00223	F	0.3	0.3	28.04	27.62	Atrophic optic nerve	Atrophic optic nerve	−13.5	−2.5	−14.75	−13	−0.5	−13.25
OFT-00092 *	M	0.1	0.05	NA	NA	Healthy retina	Peripheral toxoplasma scar	−0.5	−1.5	−1.25	−2.25	−0.75	−2.6
OFT-00097	M	0.4	0.2	26.84	26.47	Tessellated fundus, Healthy optic nerve	Tessellated fundus, Healthy optic nerve	−9.75	−5.25	−12.35	−10	−5.25	−12.6
OFT-00045	M	0.05	1	23.56	23.43	Hypopigmented fundus, Foveal hypoplasia, Colobomatous optic nerve	Hypopigmented fundus, Foveal hypoplasia	−9.75	−2.5	−11	−10	−3	−11.5
OFT-00275	F	0.7	0.1	27.61	27.6	Diffuse chorioretinal atrophy	Diffuse chorioretinal atrophy	−11.5	−1	−12	−12	−3.25	−13.6
OFT-00332	M	0.25	0.3	29.41	29.02	Tessellated fundus, Epiretinal fibrosis	Tessellated fundus, WWP on inferior and temporal retina	−15.25	−1	−15.75	−14.75	−0.5	−15
OFT-00343	F	0.8	0.8	NA	NA	Diffuse chorioretinal atrophy, Peripapillary atrophy	Diffuse chorioretinal atrophy, Peripapillary atrophy	−15.75	−2.75	−17.1	−16	−1.25	−16.75
OFT-00191	M	0.5	0.5	26.05	26.15	Diffuse chorioretinal atrophy, Mild optic nerve pallor	Diffuse chorioretinal atrophy, Mild optic nerve pallor	−9	−2	−10	−8.75	−3.25	−10.4
OFT-00391	M	0.9	NA	NA	NA	Healthy retina, WWP inferotemporal	Healthy retina	−7.25	−2.25	−8.375	−7	−3	−8.5
OFT-00407	M	0.6	0.5	28.26	27.8	Diffuse chorioretinal atrophy, Mild optic nerve pallor	Diffuse chorioretinal atrophy, Mild optic nerve pallor	−9.75	−3.5	−11.5	−9.5	−2.5	−10.75
OFT-00429	M	0.8	0.6	NA	NA	Diffuse chorioretinal atrophy, Peripapillary atrophy, WWP inferiorly	Diffuse chorioretinal atrophy, Peripapillary atrophy, WWP inferiorly	−20	0	−20	−19	0	−19
OFT-00436	M	0.63	0.3	27.42	30.93	Diffuse chorioretinal atrophy	Diffuse chorioretinal atrophy	−7	−2.5	−8.25	−15	−3.75	−16.875
OFT-00453	F	NFx	Fx	NA	NA	Complete retinal detachment	Diffuse chorioretinal atrophy, Peripapillary atrophy	2.25	−2.25	1.125	−9.5	−1.5	−10.25
OFT-00463	F	0.3	0.05	32.44	33.57	Severe peripapillary and macular atrophy	Severe peripapillary and macular atrophy	NA	NA	NA	NA	NA	NA
OFT-00474	M	0.1	0.7	27.43	25.99	Diffuse chorioretinal atrophy	Diffuse chorioretinal atrophy	−11.5	−1.25	−12.125	−10.25	−0.5	−10.5
OFT-00490	F	NA	NA	NA	NA	NA	NA	NA	NA	NA	NA	NA	NA
OFT-00506	F	0.7	0.7	NA	NA	Tessellated fundus	Tessellated fundus	−13.25	−2	−14.25	−12.5	−1.5	−13.25
OFT-00533	F	NA	NA	NA	NA	NA	NA	NA	NA	NA	NA	NA	NA
OFT-00546	M	1	1	24.45	24.12	Healthy retina	Healthy retina	−7.25	−0.75	−7.625	−5	−1	−5.5
OFT-00554	M	0.3	0.5	NA	NA	Healthy retina	Healthy retina	−9	−2.5	−10.25	−7.5	−1.25	−8.125
OFT-00559	M	0.4	0.3	NA	NA	Diffuse increase in vascular ramification, Avascular peripheral retina	Avascular peripheral retina	−7	−1.25	−7.625	−7.5	−1.75	−8.375
OFT-00568	F	0.16	0.8	34.09	33.89	Diffuse chorioretinal atrophy, Peripapillary atrophy, Staphyloma	Diffuse chorioretinal atrophy, Peripapillary atrophy, Staphyloma	NA	NA	NA	NA	NA	NA
OFT-00586	F	0.8	0.05	NA	NA	Diffuse chorioretinal atrophy	Diffuse chorioretinal atrophy	NA	NA	NA	NA	NA	NA
OFT-00590	F	0.63	0.5	29.39	28.66	Diffuse chorioretinal atrophy	Diffuse chorioretinal atrophy	−18.25	−0.25	−18.37	−20.5	−1.25	−1.75
OFT-00601	M	1	0.9	27.92	27.68	Healthy retina	Healthy retina	−7	−1	−7.5	−6.5	−1	−7
OFT-00630	F	0.08	0.08	NA	NA	Diffuse chorioretinal atrophy, Central staphyloma	Diffuse chorioretinal atrophy, Central staphyloma	−26	NA	NA	−26	NA	NA
OFT-00493	F	NA	NA	NA	NA	NA	NA	NA	NA	NA	NA	NA	NA
OFT-00175	M	0.9	0.8	31.22	30.98	Diffuse chorioretinal atrophy, Peripapillary atrophy	Diffuse chorioretinal atrophy, Peripapillary atrophy	−13.5	−4.75	−15.87	−13.25	−6	−16.25
OFT-00220	M	1	0.8	25.7	26.2	Healthy retina	Healthy retina	−5	−0.5	−5.25	−6	−2.5	−7.25
OFT-00253	F	0.9	0.9	29.59	29.1	Healthy retina	Healthy retina	−19.25	0	−19.25	−17.25	−0.5	−17.5
OFT-00268	M	0.5	0.6	27.08	27.18	Diffuse chorioretinal atrophy	Diffuse chorioretinal atrophy	−7.25	−0.75	−7.6	−7	−1	−7.5
OFT-00435	F	0.5	0.5	24.99	28.13	Healthy retina	Healthy retina	−21.25	−1.5	−22	−14	−3.25	−15.625
OFT-00443	M	1.2	1	NA	NA	RPE hypertrophy, WWP temporal and superior	Healthy retina	−7.75	−0.5	−8	−8.5	−0.5	−8.75
OFT-00477	F	1.25	1.25	NA	NA	Healthy retina	Healthy retina	−7.5	−0.75	−7.875	−8	−1.5	−8.75
OFT-00517	F	1	0.8	NA	NA	Tessellated fundus	Tessellated fundus	−6	−1.25	−6.625	−5	−2	−6
OFT-00529 *	F	NLP	0.63	NA	NA	Diffuse atrophy, Previous RD	Diffuse	NA	NA	NA	−0.25	−1.75	−1.25
OFT-00623	F	0.5	0.67	NA	NA	Tessellated fundus	Tessellated fundus	−6	−2	−7	−2.75	−0.75	−3.5

M, male; F, female; BCVA, best-corrected visual acuity; NFx, non-fixation; Fx, fixation; NLP, no light perception; OD, right eye; OS, left eye; AL, axial length; WWP, white without pressure; RD, retinal detachment; SPcc, sphere with cycloplegia; Astig, astigmatism; SE, spherical equivalent; NA, not nvailable. * Patients from family OFT-00092 and OFT-00529 had already undertaken cataract surgery and therefore did not have a higher degree of myopia in the last examination. However, the degree of preoperative myopia met the inclusion criteria.

**Table 3 ijms-23-04233-t003:** Genetic Results of Probands.

Family ID	First Diagnosis	Second Diagnosis	Gene	Transcript	Mutation	ACMG Criteria *	ACMG Result	Variant Type	Zygosity	Inheritance	Segregation Analysis Performed	De Novo /Inherited	Reported by
OFT-00074	EoHM	NYSTAGMUS AND ESOTROPIA	*TRPM1*	NM_002420.5	Allele 1: c.3121C>T: p.Arg1041Trp Allele 2: c.1023+1G>A	PM2/PVS1, PM2, PP3, PP5	VUS/P	Allele 1: Missense Allele 2: Splicing	Compound Hetero	AR	Yes	Allele 1: Maternal/Allele 2: Unknown	Allele 1: Novel/Allele 2: Miraldi Utz et al., 2018 [26]
OFT-00155	EoHM	NYSTAGMUS	*GPR143*	NM_000273.2	c.1157G>A: p.Ser386Asn	PM2, PP1, PP2	VUS	Missense	Hemi	X-linked	Yes	Maternal	Novel **
			*CACNA1F*	NM_005183.3	c.2924G>A: p.Arg975Gln	PM2, PP1, PP3	VUS	Missense	Hemi	X-linked	Yes	Maternal	Novel **
OFT-00209	EoHM	-	*TIMP2*	NM_003255.5	c.498C>G: p.Ile166Met	PM2, PP3	VUS	Missense	Hetero	AD	Yes	Maternal	Novel **
			*COL9A1*	NM_001851.6	c.6G>T: p.Lys2Asn	PM2	VUS	Missense	Hetero	AD	Yes	Unknown	Novel **
OFT-00177	EoHM	CONE-ROD DYSTROPHY AND SUBCAPSULAR CATARACT	*CEP290*	NM_025114.4	c.5777G>C: p.Arg1926Pro	PM2, PP3	VUS	Missense	Hetero	S	No	Unknown	Wiszniewski et al., 2011 [27]; Sheck et al., 2018 [28]; Sallum et al., 2020 [29]
			*PCDH15*	NM_001142763.2	c.5308_5313del: p.Ala1770_Pro1771del	PM2, PM4, PP3	VUS	Deletion	Hetero	S	No	Unknown	Novel **
OFT-00178	EoHM	-	*LRP5*	NM_002335.4	c.4610C>T: p.Ala1537Val	PM2	VUS	Missense	Hetero	AD	No	Paternal	Novel **
OFT-00181	EoHM	RETINAL DYSTROPHY	*COL2A1*	NM_001844.5	c.2818C>T: p.Arg940Ter	PVS1, PP5, PM2, PP3	P	Nonsense	Hetero	AD	Yes	Maternal	Kondo et al., 2016 [30]; Maddirevula et al., 2018 [31]; Zhou et al., 2018 [4]
OFT-00223	EoHM	-	*PEX1*	NM_000466.3	c.440T>C: p.Val147Ala	PM2, PP3	VUS	Missense	Hetero	AD	Yes	Maternal	Novel **
			*VDR*	NM_001017536.2	c.1223G>A: p.Arg408His	PM1, PM2	VUS	Missense	Hetero	AD	No	Maternal	Novel **
			*MMP9*	NM_004994.3	c.822G>C: p.Glu274Asp	PM2	VUS	Missense	Hetero	AD	Yes	Paternal	Novel **
OFT-00092	EoHM	RETINAL DYSTROPHY	*KCNV2*	NM_133497.4	c.458G>A: p.Arg153His	PM2, PP3	VUS	Missense	Hetero	S	No	Unknown	Novel **
OFT-00097	EoHM	NYSTAGMUS AND ASTIGMATISM	*CFH*	NM_000186.4	c.907C>T: p.Arg303Trp	PM2, BP4	VUS	Missense	Hetero	S	No	Unknown	Novel **
			*CACNA1F*	NM_001256789.3	c.4471C>T: p.Arg1491Ter	PVS1, PP5, PM2, PP3	P	Nonsense	Hemi	AD	No	Maternal	Novel **
OFT-00045	EoHM	NYSTAGMUS AND RETINAL DYSTROPHY	*PAX6*	NM_001258462.3	c.262A>G: p.Ser88Gly	PS2, PM1, PM2, PP2, PP3	P	Missense	Hetero	AD	Yes	De novo	Novel **
OFT-00275	EoHM	-	*COL2A1*	NM_001844.5	c.1783delC: p.Ala595LeufsTer34	PVS1, PS2, PM2, PP3	P	Frameshift	Hetero	AD	Yes	De novo	Novel **
OFT-00332	EoHM	-	*ZNF644*	NM_201269.3	c.1366A>T: p.Thr456Ser	PM2	VUS	Missense	Hetero	AD	Yes	Maternal	Novel **
			*CRYBB3*	NM_004076.5	c.547G>T: p.Glu183 *	PM2, PP3	VUS	Nonsense	Hetero	AD	Yes	Maternal	Novel **
			*LRP5*	NM_002335.4	c.263A>G: p.Lys88Arg	PM2	VUS	Missense	Hetero	AD	Yes	Maternal	Novel **
OFT-00343	EoHM	-	*OPA1*	NM_130837.3	c.1294G>A: p.Val432Ile	PM1, PM2, PP2, PP3, PP5	LP	Missense	Hetero	AD	Yes	Paternal	Stewart et al., 2008 [32]; Yu-Wai-Man et al., 2011 [33]
OFT-00191	EoHM	-	*COL11A1*	NM_001854.4	c.2900G>T: p.Gly967Val	PM2, PP1, PP3	LP	Missense	Hetero	AD	Yes	Paternal	Novel **
OFT-00391	EoHM	ASTIGMATISM	*CRYGC*	NM_020989.4	c.179G>A: p.Arg60Gln	PM2, PP2	VUS	Missense	Hetero	S	No	Unknown	Novel **
			*RDH5*	NM_001199771.2	c.683G>A: p.Arg228Gln	PM2, PP2, BP4	VUS	Missense	Hetero	S	No	Unknown	Novel **
OFT-00407	EoHM	CONE-ROD DYSTROPHY	*ARL6*	NM_177976.3	c.362G>A: p.Arg121His	PM2, PM3, PP2, PP3, PP5	LP	Missense	Homo	AR	Yes	Maternal and Paternal	Patel et al., 2016 [34]; Abouelhoda et al., 2016 [35]
OFT-00429	EoHM	-	*MMP9*	NM_004994.3	c.1270C>A: p.Arg424Ser	PM2, BP4	VUS	Missense	Hetero	S	No	Unknown	Novel **
			*IGF1R*	NM_000875.5	c.3784A>C: p.Ile1262Leu	PM2, PP3	VUS	Missense	Hetero	S	No	Unknown	Novel **
OFT-00436	EoHM	-	*MMP10*	NM_002425.3	c.497-2A>G	PP3, BS1	VUS	Splicing	Hetero	AD	Yes	Maternal	Novel **
OFT-00453	EoHM	RETINAL DYSTROPHY AND PERSISTENT FETAL VASCULATURE RIGHT EYE	*COL2A1*	NM_001844.5	c.157C>T: p.Arg53Trp	PM2, PP2, PP3	VUS	Missense	Hetero	S	No	Unknown	Novel **
			*TRPM1*	NM_001252020.1	c.3618C>G: p.Phe1206Leu	PM2	VUS	Missense	Hetero	S	No	Unknown	Novel **
OFT-00463	EoHM	-	*EPHA2*	NM_004431.5	c.308G>A: p.Arg103His	PM2, PP3	VUS	Missense	Hetero	S	No	Unknown	Novel **
OFT-00474	EoHM	-	*MERTK*	NM_006343.3	c.2264G>A: p.Arg755His	PM2, PP3	VUS	Missense	Hetero	AR	Yes	Maternal	Novel **
OFT-00490	EoHM	-	*COL11A1*	NM_001854.4	c.1021G>C: p.Glu341Gln	PM2	VUS	Missense	Hetero	S	No	Unknown	Novel **
			*TRPM1*	NM_001252020.1	c.4550C>T: p.Thr1517Met	PM2, BP4	VUS	Missense	Hetero	S	No	Unknown	Novel **
OFT-00493	EoHM	-	*CRYGA*	NM_014617.4	c.287A>G: p.Asp96Gly	PM2, BP4	VUS	Missense	Hetero	S	No	Unknown	Novel **
OFT-00506	EoHM	-	*ZNRF3*	NM_001206998.2	c.2221G>A: p.Glu741Lys	PM2, BP4	VUS	Missense	Hetero	AD	Yes	Unknown	Novel **
OFT-00533	EoHM	-	*SCO2*	NM_001169111.1	c.334C>T: p.Arg112Trp	PM2, PP5	VUS	Missense	Hetero	S	No	Unknown	Jiang et al., 2015 [36]
OFT-00546	EoHM	-	*LAMA2*	NM_000426.4	c.6880G>T: p.Val2294Leu	PM2, PP3	VUS	Missense	Hetero	AR	Yes	Paternal	Novel **
OFT-00554	EoHM	-	*SCO2*	NM_001169111.1	c.341G>A: p.Arg114His	PS3, PM2, PP3, PP5	LP	Missense	Hetero	AD	Yes	Maternal or Paternal	Tran-Viet et al., 2013 [37]; Pacheu-Grau et al., 2015 [38]; Kars et al., 2021 [39]
OFT-00559	EoHM	NYSTAGMUS	*NDP*	NM_000266.4	c.313_314delGCinsTT: p.Ala105Leu	PM1, PM2, PM5, PS1, PP2, PP3	P	Deletion/Insertion	Hemi	S	No	Unknown	Novel **
OFT-00568	EoHM	-	*PEX1*	NM_000466.3	c.3250A>G: p.Met1084Val	PM2	VUS	Missense	Hetero	S	No	Unknown	Novel **
OFT-00586	EoHM	RETINAL DYSTROPHY LEFT EYE	*MMP1*	NM_002421.4	c.1389G>A: p.Trp463Ter	PP3, BS1	VUS	Nonsense	Hetero	S	No	Unknown	Novel **
OFT-00590	EoHM	-	*COL11A1*	NM_001854.4	c.1570C>T: p.Arg524Trp	PM2, PP3	LP	Missense	Hetero	S	No	Unknown	Novel **
OFT-00601	EoHM	-	*GPR143*	NM_000273.3	c.47C>A: p.Ala16Glu	PM2, PP2, PP3, BP6	VUS	Missense	Hemi	X-linked	Yes	Maternal	Novel **
OFT-00630	EoHM	-	*CRYBA1*	NM_005208.5	c.190C>T: p.Arg64Trp	PM2, PP3	VUS	Missense	Hetero	S	No	Unknown	Novel **

P, pathogenic; LP, likely pathogenic; VUS, variants of unknown significance; Hemi, hemizygous; Hetero, heterozygous; Homo, homozygous; AR, autosomal recessive; R, recessive; AD, autosomal dominant; S, sporadic; ACMG, American College of Medical Genetics and Genomics. * ACMG Criteria in Appendix B, Table A1/** Not previously reported in the literature.

## Data Availability

Not applicable.

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
