# Peer review of "Next-Generation Sequencing Screening of 43 Families with Non-Syndromic Early-Onset High Myopia: A Clinical and Genetic Study"

_ijms, 2022, doi:10.3390/ijms23084233_

Round 1

Reviewer 1 Report

The authors addressed all suggested points in previous submission, so I consider the manuscript suitable for publication. Only a language check should be performed to correct several mistakes.

Reviewer 2 Report

The manuscript entitled “Next-Generation Sequencing Screening of 43 Families with Non-Syndromic Early-Onset High Myopia: Clinical and Genetic Study” deals with a genetic study of the Early-Onset High Myopia. The authors included a total of 43 patients performed a mutational analysis of Spanish patients diagnosed with Early-Onset High Myopia implementing a customized NGS panel containing 419 genes related to ophthalmological disorders with a suspected genetic cause.

Overall, the manuscript is well written and is easy to understand. The references used in the manuscript are recent and are adequate.  Regarding the novelty of the manuscript, it provides new insights on the genetic factor of the Early-Onset High Myopia, and as far as I know this is the first study on the Spanish population. The experiments carried out were enough and suitable for the purpose of the manuscript.

In my opinion, the results shown in this manuscript are interesting for a broader community.

Best regards

This manuscript is a resubmission of an earlier submission. The following is a list of the peer review reports and author responses from that submission.

Round 1

Reviewer 1 Report

González-Iglesias et al. produced a very interesting article describing the “Next-Generation Sequencing Screening of 43 Families with Non-Syndromic Early-Onset High Myopia: Clinical and Genetic Study”. I consider the manuscript very fascinating but, at the same time, I suggest several revisions needed to improve the reliability and the completeness of the paper:

  • Materials and Methods: this section should be really improved. I suggest to work on several points:
    • Why did the authors used hg19 rather than hg38 as reference genome?
    • The “Genetic analysis” should be divided into several sub-chapters, e.g. “Bioinformatic analysis”
    • The methodology lacks a defined statistical analysis, e.g. for association between variants and phenotypes
    • Was the experiment performed at least in triplicates?
  • The “Discussion” chapter is not sufficiently updated. I suggest the authors to add more recent references related to inflammation, oxidative stress, angiogenesis and genes came out from the study or at least present in the used OFT v.3.1 panel. The recent PMID: 34440511, PMID: 34058230 and PMID: 33801777 could represent a substrate able to enforce the role of considered genes.
  • Finally, manuscript requires English revisions and typos correction.

Reviewer 2 Report

This manuscript tried to identify genetic markers for early-onset high myopia using NGS. This is important since early-onset high myopia may eventually lead to blindness.  However, this manuscript is still too premature to be published. 

  1. extensive English editing is required. For example, line 37 (abstract). "We defend systematic genetic..." I completely do not understand why use "defend". Line 80,81: "in the cases were a NGS panel...." what cases? Line 129: "Positive family history was found in 23.3%.." What kind of  family history? There are many confusing sentences throughout the entire manuscript.
  2. How many patients were sequenced? Line 101 mentioned 43 patients from 43 families; in Line 574 49 patients from 43 families. And more importantly, the authors also mentioned some of the parents were also sequenced. How come it is only 49 individuals (43 families) were sequenced? 
  3. Where is the control group? How can the authors define the mutation or gene locus without a control group?